



# Inherent optical properties and optical characteristics of dissolved organic and particulate matter in an Arctic fjord (Storfjorden, Svalbard) in early summer

Tristan Petit[1,2*], Børge Hamre[2], Håkon Sandven[2], Rüdiger Röttgers[3], Piotr Kowalczuk[4], Monika Zablocka[4] and Mats A. Granskog[1]

[1] Norwegian Polar Institute, Fram Centre, Tromsø, Norway
[2] University of Bergen, Institute of Physics and Technology, Bergen, Norway
[3] Helmholtz-Zentrum Hereon, Institute of Coastal Ocean Dynamics, Geesthacht, Germany
[4] Institute of Oceanology, Polish Academy of Sciences, Sopot, Poland

*Correspondence to*: Tristan Petit (tristan.petit@npolar.no)

**Abstract.** There have been considerable efforts to understand the hydrography of the Storfjorden fjord (Svalbard). A recurring winter polynya with large sea ice production makes it an important region of dense water formation at the scale of the Arctic Ocean. In addition, this fjord is seasonally influenced by freshwater inputs from sea-ice melt and the surrounding islands of the Svalbard archipelago which impacts the hydrography. However, the understanding of factors controlling the optical properties of the waters in Storfjorden are lacking and are crucial for development of more accurate regional bio-optical models. Here, we present results from the first detailed optical field survey of Storfjorden conducted in early summer of 2020. In addition to the expected seasonal contribution from phytoplankton, we find that in early summer waters in Storfjorden are optically complex with a significant contribution from coloured dissolved organic matter (33–64% of the non-water absorption at 443 nm) despite relatively low CDOM concentrations, and in the nearshore or near seabed from non-algal particles (up to 61% of the non-water absorption at 550 nm). In surface waters, the spatial variability of light attenuation was mainly controlled by inorganic suspended matter originating from river runoff. A distinct subsurface maximum of light attenuation was largely driven by a subsurface phytoplankton bloom, controlled by stratification resulting from sea-ice melt. Lastly, the cold dense bottom waters of Storfjorden, from winter sea ice production, which periodically overflows into the Fram Strait, was found to contain elevated levels of both non-algal particles and dissolved organic matter, which is likely caused by the dense flows of the nepheloid layer interacting with the sea bed.

## 1 Introduction

Located in the northern Barents Sea, Storfjorden, the largest fjord in the Svalbard archipelago, is influenced by two major water masses, namely warm Atlantic Water (AW) from the Norwegian Atlantic Current and cold Arctic Water (ArW), which is AW cooled along the traverse around Spitsbergen entering the fjord from the north and east with the East Spitsbergen Current (ESC). The recurring winter polynya in the Storfjorden is considered an important "ice factory" and winter ice formation here supports production of about 5–10% of the dense water formed in the whole Arctic Ocean (Smedsrud et al.,



2006). This is related to subsequent release of salty brine from sea ice to the underlying relatively shallow waters which creates favourable conditions for formation of cold salty brine-enriched shelf waters (BSW) of high density (Skogseth et al.,

2005). While these phenomena are well documented, mainly from hydrographical surveys, knowledge of the Inherent Optical Properties (IOPs) of waters in Storfjorden is lacking, which holds back the development of regional bio-optical models.

At a larger scale, over the Arctic Ocean, the largest recent reduction in the extent of sea ice in winter has been found to take

place in the Barents Sea (Onarheim et al., 2018). While typically the Barents Sea is completely ice-free in summer and early to late autumn, a reduced winter/spring ice cover can have a significant impact on the light climate and the phenology of associated ice algae and phytoplankton (Leu et al., 2011). Recent studies have shown a northward progression of the polar front with associated changes to the seasonal plankton bloom and the sea ice reduction in the Barents Sea (Neukermans et al., 2018; Oziel et al., 2017). As sea ice continues to retreat, understanding the evolution of the optical properties of the water

column, and in particular IOPs, is crucial for better understanding of the potential changes for phytoplankton dynamics.

The two complementary ways of getting IOP observations in oceanic waters are (*i*) passive multi-spectral satellite imagery in the visible, often referred to as "Ocean Colour" imagery, and (*ii*) field-based measurements. While the former theoretically enables getting better spatial and temporal information on the IOPs dynamics, it is especially difficult to operate in polar

regions (International Ocean Colour Coordinating Group (IOCCG), 2015). Despite their sparse nature, field observations thus remain the main source of information regarding IOPs in the Arctic. They also constitute a unique way of capturing the usually complex vertical structure of the optical properties in Arctic waters, whose knowledge is essential for the remote sensing of stratified waters (Lee et al., 2020). In addition,  increasing the number of observations through new surveys is critical for strengthening the statistical characterization of the parameters used when building regional bio-optical models

(eg. Kostakis et al., 2020), both for calibration/validation of ocean colour satellite products (eg. Orkney et al., 2020) and for getting a correct representation of optical properties in numerical models.

While we are not aware of any existing optical surveys in the Storfjorden, there have been a few studies examining the optical characteristics of waters in the Barents Sea proper (Aas & Berge, 1976; Aas & Høkedal, 1996; Falk-Petersen et al.,

2000; Hancke et al., 2014; Hovland et al., 2014; Kostakis et al., 2020; Orkney et al., 2020). Further, a number of other studies have been conducted in the region of AW inflow west and north of Svalbard (Kowalczuk et al., 2019; Makarewicz et al., 2018; Pavlov et al., 2015 2017). All these studies point toward a significant contribution of phytoplankton to light attenuation in the open Barents Sea both north and south of the polar front, since the concentration of coloured dissolved organic matter (CDOM) in water masses of AW origin is relatively low, compared to 'true' polar waters with a distinctly

higher CDOM signal (Pavlov et al., 2015). The cold ArW entering the Barents Sea from the north, and located north of the polar front, is also a low-CDOM water (Hancke et al., 2014), likely modified AW that has been cooled and somewhat





freshened on its traverse around Svalbard, since similar low-CDOM ArW is found north of Spitsbergen (Kowalczuk et al., 2017; Pavlov et al., 2015). On the contrary, 'true' polar waters carry a distinct terrestrial CDOM signal from Arctic river runoff, with much higher CDOM concentrations as found in the East Greenland Current at the same latitude as Storfjorden

(Granskog et al., 2012; Pavlov et al., 2015). Local runoff and meltwaters from land and glaciers are potential sources of optically active constituents, both particulate and dissolved matter, as evidenced in several Spitsbergen fjords (Halbach et al., 2019; McGovern et al., 2020; Pavlov et al., 2019; Sagan & Darecki, 2018), where turbidity induced by particles from river runoff and glacial melt waters often control the light climate and thus productivity instead of phytoplankton or CDOM (Halbach et al., 2019).


Given the complex dynamics in Storfjorden, and the vicinity of the surrounding land masses we surmise that factors affecting light attenuation in the Storfjorden are likely driven by a complex interplay of the local circulation and dynamics, biological production in the fjord, and is, to some degree, influenced seasonally by local runoff as in other Spitsbergen fjords. Further, the dense BSW formed in winter may also carry some higher turbidity (see e.g. (Bensi et al., 2019)), when

these dense waters flow along the seafloor to their density-determined depths. In order to assess these assumptions, we here report on the first survey of the IOPs in Storfjorden shortly after sea ice has disappeared, since access to the fjord is very challenging in the ice season due to very dynamic ice conditions. Based on our field observations, we present a detailed characterization of the three types of optically active substances, namely phytoplankton, CDOM, and non-algal particles. We provide new insight of the factors controlling light attenuation in this fjord in early summer, as well as on the relative

contribution and origin of these substances.

## 2 Materials and Methods

The field work took place between 20–30 June 2020 during an expedition on the Norwegian coast guard vessel *KV Svalbard*, that was undertaken as part of the *Useful Arctic Knowledge Research School,* hereafter UAK2020 (Sagen et al., 2020).

### 2.1 Conditions and sampling stations

At the time of the UAK2020 expedition there was no (or very little remnants of) sea ice in the fjord, and based on operational ice charts (https://cryo.met.no) the sea ice had completely disappeared from the fjord only some days before the expedition reached the area. The layout of the sampling program was made to cover a section from the southern part of the fjord (Storfjordrenna) to the northernmost inner part of the fjord, crossing a prominent sill (sill depth about 120 m while the bottom depth of the deepest part of the inner fjord is ~180 m). In addition, a section at the latitude of the sill was conducted

across the fjord. Optical measurements and sampling were undertaken at the stations shown in Figure 1, while extra hydrographical casts were conducted in addition to these stations (positions not shown). This study only focuses on the South–North transect for sake of brevity, but the entire optical dataset is available (Petit et al., 2021).

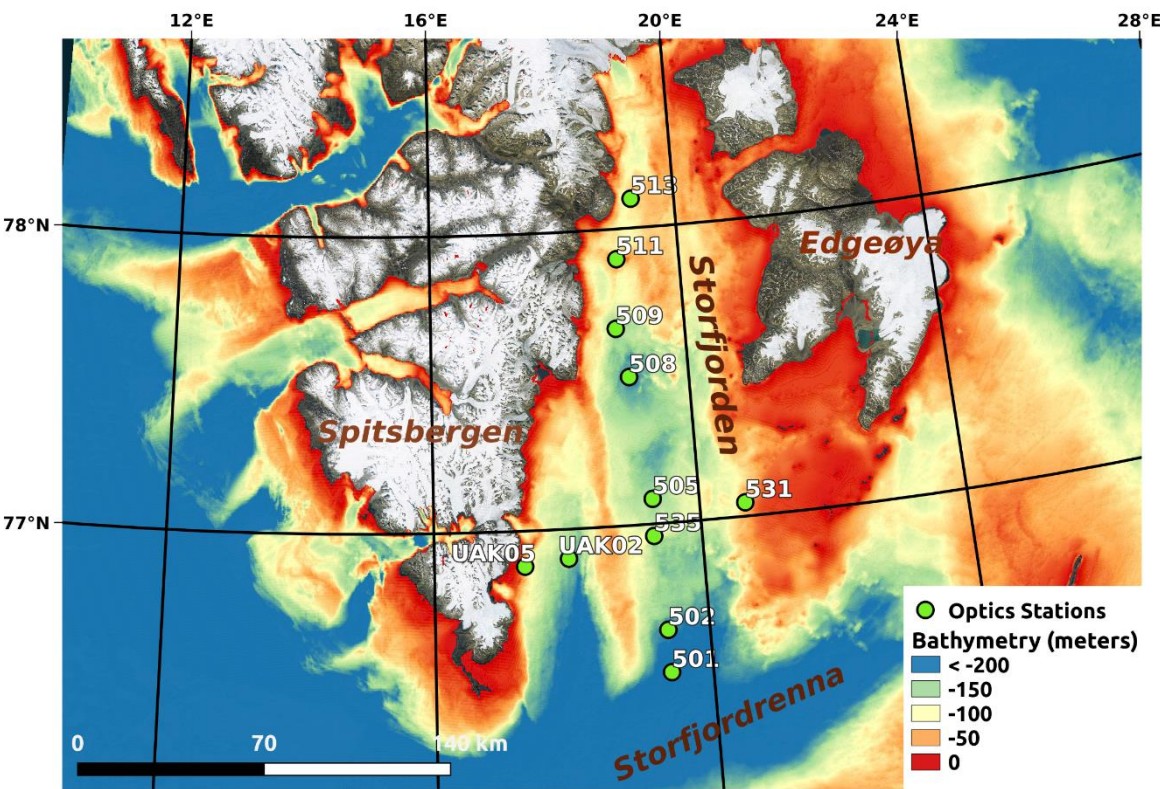

**Figure 1- Stations visited for optical measurements during the UAK2020 cruise in Storfjorden (Svalbard) along with bathymetry**
**showed between 0–200 m (red to blue colours). This included seven stations along a South – North transect, from deep (>250 m)**
**waters at the entrance of the fjord, to shallow (<50 m) area in the inner fjord. Four stations across the fjord entrance (along an**
**East – West transect) were located at the latitude of the sill. Bathymetry is from GEBCO. Svalbard land part comes from Bing**
**aerial images (© Microsoft). The map projection is UTM33N.**

## 2.2 In situ measurements

Vertical profiles of IOPs and fluorescence by dissolved organic matter (FDOM), together with conductivity (Salinity, $S_p$ in

practical salinity scale), temperature, and depth (pressure) were measured at all stations from the surface down to close to the

bottom using an instrument package consisting of an ac-s attenuation and absorption meter (WET Labs Inc., USA), a

WETStar FDOM fluorometer (WET Labs Inc., USA), and a factory-calibrated SBE37 SIP conductivity-temperature-depth

(CTD) probe (Sea-Bird Electronics, USA). The data from all three instruments were stored by a DH4 data logger (WET

Labs Inc., USA). Since there was no real-time pressure reading from the optical instrument package, the maximum depth for

each cast had to be estimated using wire length. Thus, the actual distance to the bottom (from echo depth from the ship's





echosounder) varied from cast to cast. The battery failed at the two northernmost stations and only one profile down to half the bottom depth could be achieved at station 513.

Total absorption (a) and attenuation (c) were obtained from the two channels of the ac-s with a sampling frequency of 4Hz. The processing steps involved (i) blank subtraction, (ii) salinity and temperature correction and (iii) scattering correction of the absorption and attenuation channels. The blanks were measured by injecting Milli-Q water successively inside each channel using a peristatic pump with constant flow of 0.6 L/min. This was done once before and three times during cruise for tracking potential drift of the instrument. No evidence for any drift was found but we experienced more variability of the

blank performed during the cruise and thus decided to use the blanks done in laboratory before cruise for all the data processing. Salinity and temperature values were taken from the CTD data which had a sampling frequency of ~0.5Hz. Scattering correction was performed using the $flat^e$ method proposed by (Röttgers et al., 2013). This correction assumes spectral invariance of the Volume Scattering Function (VSF). It makes use of an empirical law for estimating the true absorption at 715 nm from the 715 nm a channel of the ac-s. This value is then used for correcting the whole spectrum.

Rationale behind the use of a flat method instead of a proportional is that with the latter the spectral shapes of the corrected spectra were less realistic and very low absorption spectra resulted in some (unrealistic) negative values.

FDOM was measured using a three-channel WET Labs WETStar fluorometer, with excitation/emission pairs as follows: Channel 1 (Ch1, 310/450 nm) that represents marine ultraviolet humic-like and marine humic-like material; for Channel 2

(Ch2, 280/450 nm) represents terrestrial humic-like material; and for Channel 3 (Ch3, 280/350 nm) represents protein-like tryptophane type material resulting from the presence of phytoplankton (for details see Makarewizc et al. (2018)). Fluorescence intensities acquired from the WETStar fluorometer are reported here in background-corrected raw counts (RC). The background values were estimated in lab with dark environment by creating a flow of ultra-pure (Milli-Q) water into the sensor thanks to a peristatic pump.


Water sampling was conducted with a rosette with seven 4L Niskin bottles with a SBE 19plus CTD (Seabird, Inc., USA) giving real time information about temperature and salinity. Sampling included samples for oxygen isotope ratio ($\delta^{18}O$), and samples for determining absorption by CDOM and particles (total, algal and non-algal). This sampling typically included fixed depths of surface, 10, 20, 50 m depth and a near-bottom sample.


Water samples for CDOM were collected by gravity filtration through pre-rinsed 0.2 μm Millipore Opticap XL filter capsules connected to the Niskin with silicon tubing. The samples were stored in pre-combusted amber glass vials in dark at +4 °C until analysis. CDOM samples were analysed onboard within days of collection.





Samples for particulate absorption measurements were collected on 25 mm filters (Whatman GF/F, nominal pore size 0.7 µm). Special care was taken to keep a low vacuum during filtration to avoid algae cell breakage. The filtered volume was adjusted systematically according to 2 criteria: (i) colour of the filter and (ii) filtering speed for anticipating potential clogging. Samples were immediately shock-frozen and stored in liquid nitrogen during the cruise and were later stored at - 80°C until analysis.


Samples for $\delta^{18}O$ we collected into 20 ml polyethylene bottles, filled completely to avoid any headspace, closed carefully and the caps sealed with parafilm. Samples were shipped to the Jan Veizer Stable Isotope Laboratory (University of Ottawa, Canada) for determination of the oxygen isotope ratio on a Finnigan MAT Delta plus XP + Gasbench. A precise amount of 0.6 mL of water is pipetted into Exetainer vials. No catalyst is required. The vials are flushed and filled with a gas mixture

of 2% $CO_2$ in helium. The flushed vials are left at room temperature for a minimum of 5 days. The $CO_2$ gas is analysed automatically in continuous flow. The results are normalized to VSMOW (Vienna Standard Mean Ocean Water) standard using three calibrated internal standards spanning most of the natural range. The precision of the analysis is ±0.15‰.

### 2.3 Laboratory methods

Spectral CDOM absorption coefficients were determined from measurements onboard using a Liquid Waveguide Capillary

Cell (LWCC) system. This system included a DH-mini Deuterium/Tungsten source (Ocean Optics), a 1 m long LWCC and a Flame-T spectrometer covering the UV–NIR spectral range 250–800 nm. The LWCC was cleaned with methanol at the beginning and end of each day. For each sample, three successive intensity measurements in digital counts of the dark current $I_{DC}$, reference purified water $I_{ref}$ and sample water $I_S$ were performed. The apparent absorption coefficient (in m$^{-1}$) was then determined at each wavelength by averaging the three spectra ($a_{app}$) each computed as follows:

$$a_{app}(\lambda) = \frac{\ln\left[-\left(I_s(\lambda) - I_{DC}(\lambda)\right)/\left(I_{ref}(\lambda) - I_{DC}(\lambda)\right)\right]}{l} \tag{1}$$

with $l = 1.00$ m as the optical path length. The apparent absorption coefficient of a 100 mg/l pre-burnt HPLC grade NaCl solution $a_{NaCl}$ (in m$^{-1}$) was measured with the same protocol. The salinity correction applied was as follow:

$$a(\lambda) = a_{app}(\lambda) - \frac{a_{NaCl}(\lambda)}{91} \cdot S_p \tag{2}$$

with $S_p$ the practical salinity of the sample. It should be noted that the PSU-normalized apparent NaCl absorption was obtained by dividing $a_{NaCl}$ by 91 instead of 100 for taking into account the fact that sea water includes other salts than just NaCl. This number has been determined empirically (personal communication with Rüdiger Röttgers).


The particulate absorption coefficient was quantified for each sample filter by two independent methods of the Quantitative Filter Techniques (QFT) using integrating spheres large enough for placing the filters inside them. The first one (Röttgers & Gehnke, 2012, hereafter QFT-Perkin) uses a commercial laboratory UV/VIS/NIR spectrophotometer (Lambda 950, Perkin



Elmer, USA) while the latter (Röttgers et al., 2016, hereafter QFT-ICAM) uses a custom made portable integrating cavity

absorption meter. In both cases, the optical density of the sample filters, $OD_s$, was measured against the optical density of a wet reference blank filter $OD_{ref}$, taking a dry filter as the reference in both cases. The particulate absorption $a_p$ (m⁻¹) was then determined as:

$$a_p(\lambda) = \frac{\left(OD_S(\lambda) - OD_{ref}(\lambda)\right) \cdot A \cdot \beta}{V} \qquad (3)$$

where A (m²) is the filter patch area, β = 4.5 (Röttgers & Gehnke, 2012) is the path length amplification factor and V (m³) the volume of filtered sample water. The particulate absorption coefficient was measured between 350–750 nm and 390–850 nm with the QFT-Perkin and QFT-ICAM, respectively. The custom-made QFT-ICAM showed very good agreement with the QFT-Perkin method (Figure 2). Non-Algal Particle (NAP) absorption $a_{NAP}$ (m⁻¹) was measured with the QFT-ICAM method using the same filters after bleaching them with a 1% NaOCl solution for 1–3 minutes. The bleach was removed by

oxidation using a 10% $H_2O_2$ solution, the filters stored and measured one day later. The phytoplankton absorption $a_{phy}$ (in m⁻¹) was computed as the difference of total and NAP absorption ($a_{phy}(\lambda) = a_p(\lambda) - a_{NAP}(\lambda)$).

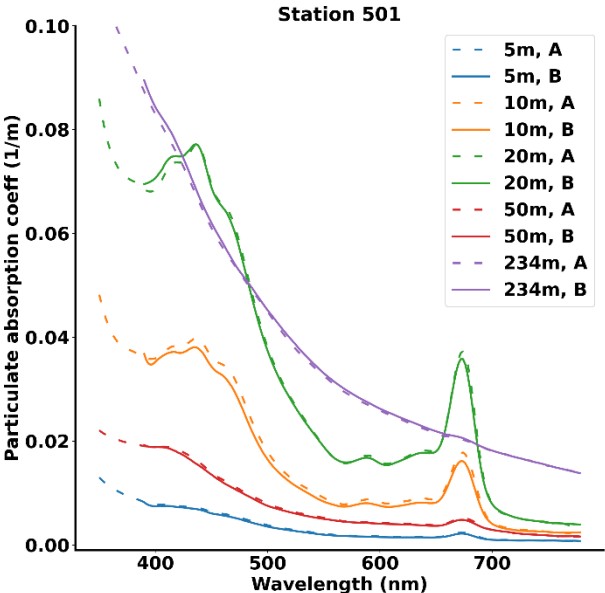

**Figure 2** - Comparison of total particulate absorption coefficient $a_p$ as derived from the QFT-Perkin (A) and the QFT-ICAM (B) methods for the southernmost station. These indicate minimal difference of the two methods for the overlapping wavelength range.





## 2.4 Satellite data

Clear-sky conditions prevailed during the days of the cruise, and we thus included as background information to this study a qualitative analysis of a Sentinel-2 image taken on 22nd June 2020 over Storfjorden. The Sentinel-2 mission, launched by the European Space Agency (ESA) under the Copernicus program, is a constellation of two satellites launched in 2015 and 2017 respectively. Each of them encompasses a MultiSpectral Imager (MSI) capturing light in 13 spectral bands from 443 nm in the visible to 2190 nm in the short-wave infrared. Compared to existing satellite missions like Sentinel3/OLCI it has a better spatial resolution (10-60 m depending on the band) which makes it suitable for catching the high spatial variability typically encountered in coastal environments.

The Sentinel-2 data was downloaded from the Copernicus Open Access Hub via the *sentinelsat* python package in geocorrected Top-Of-Atmosphere (TOA) spectral radiance (L1C product). An atmospheric correction algorithm based on Acolite (Vanhellemont & Ruddick, 2016) and implemented using the python package Py6S (Wilson, 2013) was used for converting the TOA data into a Bottom-Of-Atmosphere (BOA) reflectance. This solution uses the 6S radiative transfer model (Vermote et al., 1997) for estimating and removing the contribution of the atmospheric gases and aerosols to the remote sensing signal. The gas (resp. aerosol) model was set to subarctic summer (resp. maritime) and a manual value of 0.1 was set for the aerosol optical thickness at 440 nm (aiming at BOA reflectance around 0 in the near-infrared in areas with low turbidity). There was no swell and no wind at time of acquisition, as well as favourable solar-sensor geometry, and we thus did not have to apply any sunglint removal strategy. The algorithm of Nechad et al. (2010) was used for estimating the turbidity and enhancing the visual analysis of the surface water. It has been shown to be robust to various environmental conditions (Dogliotti et al., 2015). Considering the quasi absence of phytoplankton in surface waters due to stratification (see Sect. 3.53.5), we did not apply any chl-a retrieval algorithm to the satellite data.

## 3 Results and Discussion

### 3.1 Hydrographic setting in the Storfjorden in early summer

Section plots of salinity and temperature for the North to South transect are presented in Figure 3. A surface layer of fresher ($S_p\sim33$) and warmer ($T>2^{\circ}C$) water was found down to about 20 m depth, followed by a strong density gradient between 20-30 m depth. These section plots also highlight very dense and cold ($S_p>35.2$ and $T<-1.5^{\circ}C$) bottom waters for the stations north of the sill (at depths >125 m), but also south of the sill at the bottom. At the southernmost station in Storfjordrenna there are warmer temperatures present down to 125 m depth indicative of warmer AW from the WSC entering the Storfjordrenna. $\delta^{18}O$ values (Figure 4) were positive and ranged from 0.2–1‰ which indicate contribution from sea-ice melt at the surface and limited impact from runoff along the main north-south transect in the middle of the fjord. Only a few



northern stations with lowest $\delta^{18}O$ values at the very surface compared to other stations, show signs that local runoff could have reached the offshore location of the sampling stations.

This situation appears typical for the fjord in summer (Fossile et al., 2020), with a rather shallow fresher and warmer, well stratified surface layer present over the entire fjord, and near-freezing, salty, brine-enriched waters at the bottom of the fjord.

The surface layer was shallower than later in summer (cf. Fossile et al., 2020), likely because the sea ice had just disappeared that has limited the time for mixing and deepening of the surface layer. Below sill depth we find evidence for presence of BSW in the fjord from winter ice production. At the southernmost station in the Storfjordrenna there are signs of typical warmer AW from the West Spitsbergen Current (WSC) at intermediate depth, but also here dense, cold BSW water is present at the bottom, likely a result of an overflow of BSW over the sill from Storfjorden proper.



**Figure 3 -** Salinity (a) and temperature with white contours of density anomaly (b) along south–north transect (See Fig. 1), from the Storfjordrenna to the inner fjord. White lines indicate the location and depth of the CTD data. Plots are made with Ocean Data View (Schlitzer, Reiner, Ocean Data View, underline{odv.awi.de}, 2021).





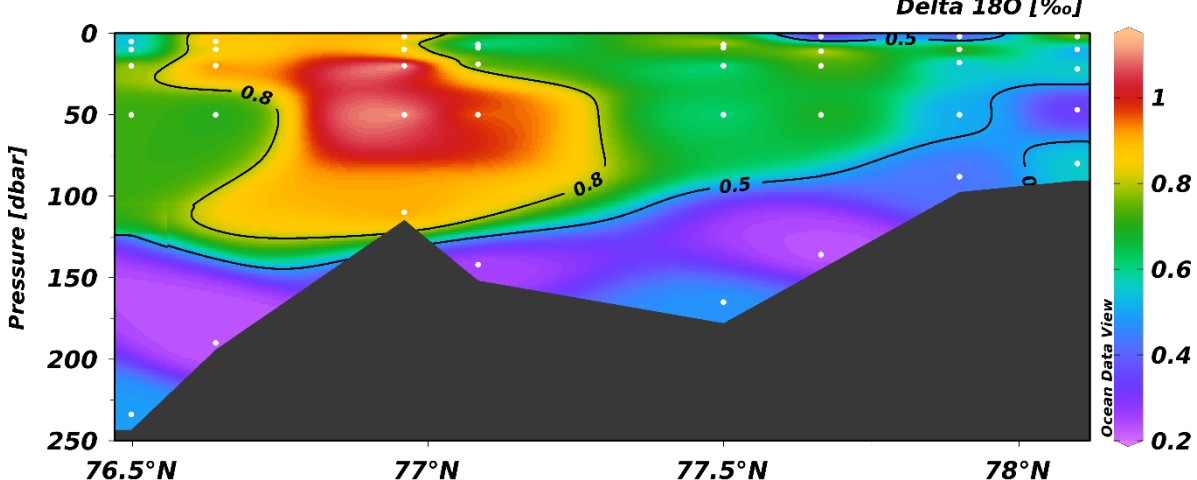

**Figure 4 - Oxygen isotope ratio (δ18O) from water samples, sample location indicated with white dots. Bathymetry is based on echo depth at the CTD stations. Plots are made with Ocean Data View (Schlitzer, Reiner, Ocean Data View, odv.awi.de, 2021).**

## 3.2 Inputs from the shore

Visual assessment of the impact of terrestrial runoff on the surface water's optical properties on 22$^{nd}$ June 2020 was performed using Sentinel-2 bottom-of-atmosphere reflectance (Figure 5, left) and turbidity (Figure 5, right). As expected from the season (late June) and associated land snow and glacier melt, significant turbidity plumes were observed nearshore with turbidity >3.5 FNU (red colour on the turbidity map) while it was <0.8 FNU (blue colour on the turbidity map) in the middle parts of the fjord. Along the main south–north transect, only the stations 509, 511 and 513, all sampled on 26$^{th}$ June (4 days after the satellite acquisition), seem to be close enough for being potentially impacted by the coastal inputs of particles and coloured dissolved organic matter. This would confirm the similar assertion made from the analysis of the $\delta^{18}O$ data.


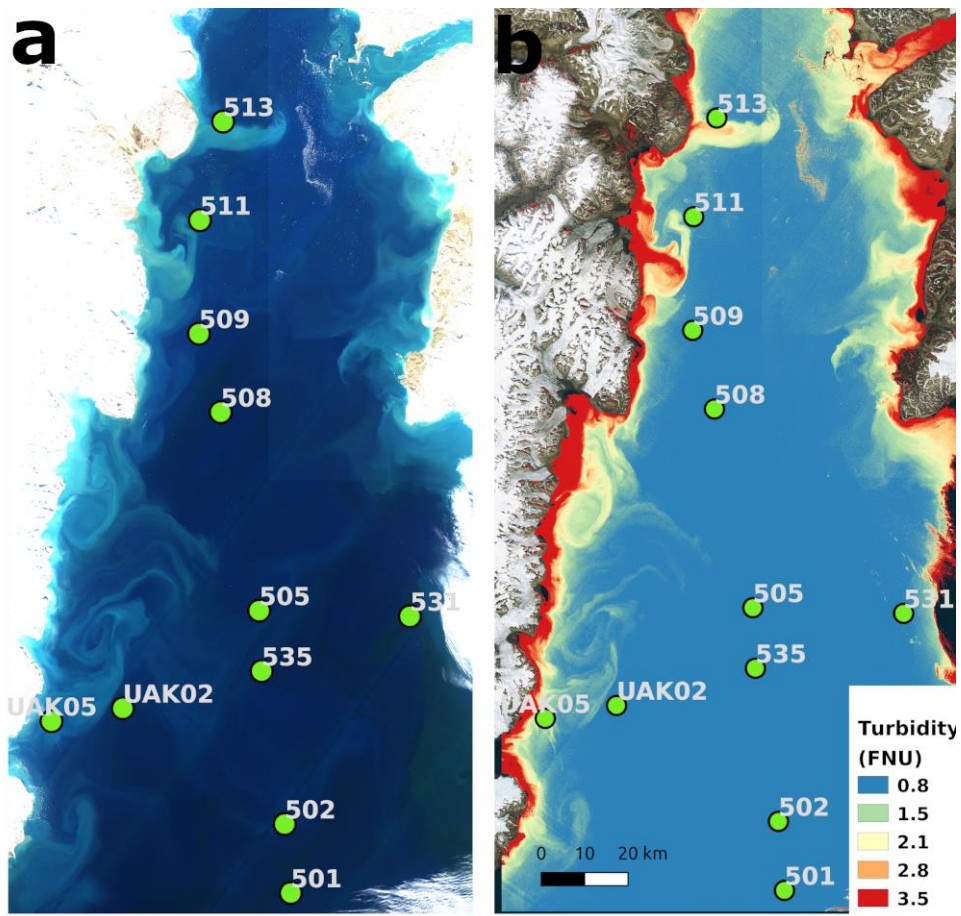

**Figure 5 - Satellite products derived from a Sentinel-2 acquisition on 22nd June 2020: (a) false colour composite of the red (665 nm), green (560 nm) and blue (492 nm) channels of the bottom-of-atmosphere (BOA) reflectance, with values shown between 0–0.07, (b) turbidity map derived from the BOA reflectance with land part for illustration coming from Bing aerial image (© Microsoft). The position of the sampling station is indicated.**

### 3.3 Observed inherent optical properties in relation to hydrography

In situ observations of the attenuation coefficient at 440 nm, $c(440)$ (Figure 6, panel a), which is the sum of absorption and scattering coefficients and varies both with CDOM, phytoplankton and non-algal particles concentration, show how IOPs are related to the stratification and water masses in the fjord and/or to the inputs from land. The ratio between the scattering and absorption at 440 nm, $b(440)/a(440)$ (Figure 6, panel b) gives some indication about the nature of the IOPs, with very high values corresponding to strong scatterers like mineral particles and lower values obtained when the IOPs are dominated by CDOM or phytoplankton.

There are distinct subsurface maxima of attenuation with very high scattering over absorption ratio (>11) for the northern most station 513. This could be linked to land inputs visible on the satellite data (Figure 5). For most of the other stations we can see distinct subsurface maxima in $c(440)$ with lower $b(440)/a(440)$ between 25–50 m depth that may be linked to



presence of phytoplankton. AW in Storfjordrenna and waters at intermediate depth in the fjord show the lowest attenuation and here also the contribution from scattering is lowest. We found elevated scattering relative to absorption at the very bottom at several stations, mostly next to the sill. Apart from station 513, we also observed widespread highly scattering particles in the surface layer (often with low concentration, as deduced from concomitant low $c(440)$). They could

potentially result from very small (fine grain size) non-algal particles melted out from sea ice (Bélanger et al., 2013; Granskog et al., 2015) or, as suggested very recently in (Davies et al., 2021), by some large non-mineral particles such as fish eggs or zooplankton. The ratio of scattering to absorption coefficient was generally very high in Storfjorden, up to 11 times more scattering than absorption at 440 nm (which corresponds to a single-scattering albedo b/c of 0.92). Similar increase in scattering was also observed west and north of Spitsbergen in surface layers affected by sea-ice melt (Granskog

et al., 2015).





**Figure 6 - Total attenuation coefficient (c) with white contours of potential density anomaly (panel a) and ratio of scattering (b) to absorption (a) at 440 nm (panel b) from in situ ac-s data for the South–North transect (see Figure 1). Plots are made with Ocean Data View (Schlitzer, Reiner, Ocean Data View, odv.awi.de, 2021).**

**3.4 Characteristics of FDOM and CDOM**

As already observed for AW in the WSC (Makarewicz et al., 2018) we found no signs of elevated concentrations of humic-like FDOM in Storfjorden (FDOM Ch1 and Ch2, not shown), nor any significant variation between stations or with depth for these channels. On the other hand, the tryptophan-like compound (Figure 7) shows distinct patterns with a subsurface maximum linked to the density gradient visible on Figure 6 (white isolines). This is linked to the subsurface chlorophyll

maxima (SCM) that we observed (see Sect. 3.5), which are common in Arctic waters (Ardyna et al., 2013). The variation of



(F)DOM in the fjord is thus largely linked to biological activity in the surface waters and tends to demonstrate low variability of the other (humic-like) sources of DOM in Storfjorden, which in turn points to a limited contribution of DOM from land runoff to the fjord's DOM pool.

CDOM, quantified here by its absorption coefficient at 440 nm $a_{CDOM}(440)$, shows values within the range 0.02–0.05 m$^{-1}$, which are levels typical for AW in the WSC (a$_{CDOM}$(443): 0.016–0.51 m$^{-1}$, (Kowalczuk et al., 2019) and in the Barents Sea proper (a$_{CDOM}$(443): 0.035–0.162 m$^{-1}$, (Hancke et al., 2014). Although the resolution of water samples was low, some distinct patterns appear. The very surface layer seems to have the lowest CDOM, which is likely linked to dilution from low-CDOM sea-ice melt (c.f. Granskog et al., 2015). The distinct subsurface peak of the tryptophan-like compound (Figure 7)

does not appear that clearly in CDOM, suggesting that at the time of sampling the impact from local phytoplankton-based production in CDOM is also limited. However, there was elevated CDOM absorption at the very bottom at several stations, linked with the dense and cold BSW. The spectral slope coefficient, $S_{350-550}$, was in the range of 0.013–0.0167nm$^{-1}$. Values of the slope coefficient observed in Storfjorden in June 2020 matched very well with observations east off Storfjorden in the Barents Sea proper (Hancke et al., 2014), but were lower compared to spectral slope values in AW in the WSC (Makarewicz

et al., 2018). The lowest spectral slope values were observed in the southern end of the transect and corresponded to inflowing AW water. Highest value of the $S_{350-550}$ >0.0165 nm$^{-1}$ were observed in the bottom waters, linked with cold and dense BSW.








**Figure 7** - Section plots of FDOM channel 3 (tryptophan-like material) of the in situ sensor (a), CDOM absorption at 440nm (b)
and the spectral slope of CDOM absorption in the wavelength range 350 to 550 nm (c) from water samples. Data locations shown
with white dots. Plots are made with Ocean Data View (Schlitzer, Reiner, Ocean Data View, odv.awi.de, 2021).

### 3.5 Characteristics of particulate matter

In situ absorption measurements also indicated that the subsurface maxima in tryptophan-like FDOM, scattering and
absorption were linked to phytoplankton biomass. The absorption line height at 676nm ($a_{LH}(676)$, Roesler & Barnard, 2013)
was computed from the in situ ac-s absorptions at 650, 676 and 715 nm as a proxy for phytoplankton biomass (Figure 8,
top). This confirmed that the subsurface maxima in tryptophan-like FDOM and attenuation at 440 nm were due to increase in
phytoplankton concentration for the central (maximum $a_{LH}(676)$ around 25 m depth) and southern (max. $a_{LH}(676)$ around
50 m depth) part of the fjord. This subsurface layer of increased phytoplankton abundance was interfacing the frontal zone
between inflowing AW and ArW. This is likely due to nutrients limitation shallower and light limitation deeper than this
zone.

Sample-based values of $a_{phy}(440)$ and $a_{NAP}(440)$ (Figure 8, middle and bottom) and their distribution with depth were in
similar range as reported in WSC waters by (Kowalczuk et al., 2017, 2019), further pointing to the presence of AW
dominated waters. Although not targeting the subsurface chl-a maxima at time of sampling (since in situ data was not
available to guide the water sampling), the sample-based phytoplankton absorption at 440 nm (Figure 8, middle) indicates
increased absorption by phytoplankton in the subsurface, and at deeper depth in the southern fjord compared to the central
part of the fjord.

The surface waters of the Storfjorden were characterised by very low phytoplankton absorption. This is also clearly visible
from the spectral absorption of particles shown for stations 501 and 508 (Figure 9), here the typical absorption peaks for
chlorophyll are evident only at subsurface depths. Slight increase in surface non-algal particle absorption was only observed
for the northern station 509 and, to a lesser extent, at station 511. This implies overall limited influence of absorbing
particles from land runoff along the North–South transect.

Non-algal particle absorption from the water samples (Figure 8, bottom) showed elevated values for the bottommost samples
(with nearly no phytoplankton absorption), suggesting that there are turbid, near-bottom flows dominated by inorganic
material, likely resuspension of bottom sediments. This increase in $a_{NAP}(440)$ is most likely due to mineral particles
deposited on the seabed and resuspended due to water flows typically encountered in the nepheloid layer. This increase may
also partly originate from both incomplete mineralization of settled phytoplankton particles or be an effect of the adsorption
of the CDOM fraction to resuspended particles. (Yamashita et al., 2021) has described the effect of enrichment in bottom
water flowing over the productive shelf of Sea of Okhotsk with allochthonous FDOM contained within sediments pore
waters. This effect could also be present in the Storfjorden, supported by our observation of significant decline of the local
aLH(676) (phytoplankton biomass proxy) maximum with depth (Figure 7, top).










**Figure 8 - Section plots of the absorption line height (aLH) at 676nm with white contours of density anomaly (panel a, data from the ac-s), absorption by phytoplankton (panel b, data from the water samples) and absorption by non-algal particles (NAP) (panel c, data from the water samples) all at 440nm. Plots are made with Ocean Data View (Schlitzer, Reiner, Ocean Data**
**View, odv.awi.de, 2021).**

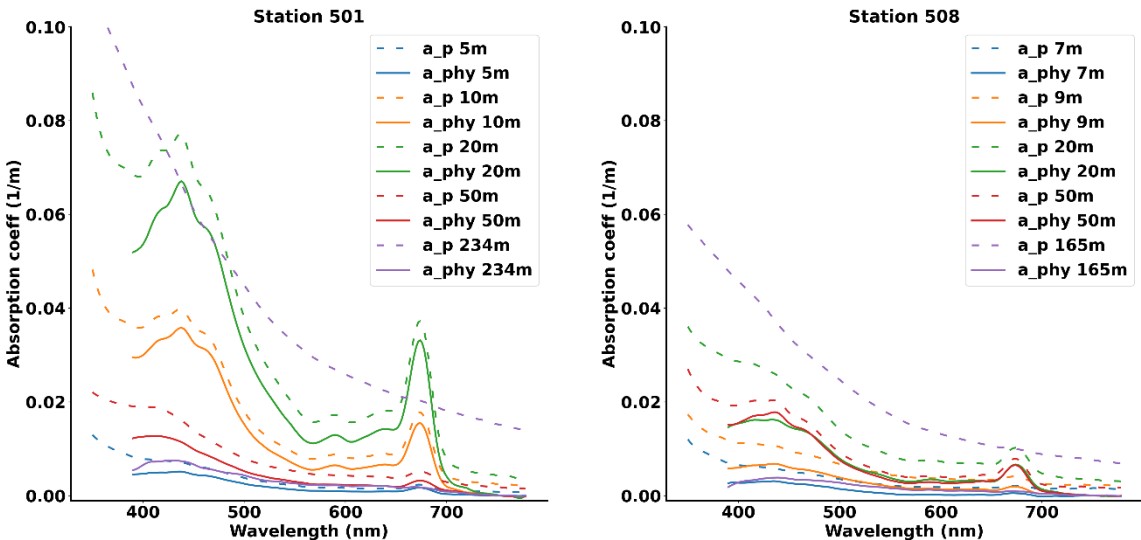

**Figure 9 - Comparison of total particulate a$_p$ (dashed line) and phytoplankton a$_{phy}$ (solid line) absorption in Storfjordrenna (station 501) and in the fjord north of the sill (station 508) with depth.**

## 3.6 Relative contribution of the optically active constituents

Getting knowledge on the relative contribution of the optically active constituents, namely phytoplankton, CDOM and non-algal particles, to the optical characteristics of sea water is of particular importance for any radiative transfer modelling or optical remote sensing application of a specific area. Considering the optical complexity found in Storfjorden, we decided to conduct separate analysis for the surface (0–15 m water depth), subsurface (15–60 m water depth) and dense bottom (<20 m from the bottom, stations 505, 508 and 509 only) waters. In addition, we further separated the surface and subsurface into

two groups: (*i*) northern stations 509, 511 and 513 which are potentially land-influenced, (*ii*) southern offshore stations 501, 502, 505, 535, and 508 which are less influenced by land runoff. For each of the five resulting categories, an absorption budget (excluding water itself) was built based on the water sample data (Figure 10) and an attenuation budget (excluding water itself) was computed based on the in situ ac-s data (Figure 11). A corresponding table with averages of relevant optical properties for each of the five categories is presented in Table 1.




The non-water IOPs in Storfjorden were driven by non-algal particles and CDOM for the Northern surface water, by CDOM for the southern surface waters, by phytoplankton in the subsurface waters and by non-algal particles and CDOM for the bottom dense water. The variability in attenuation was driven by variability of scattering (Figure 11, light blue) which was highest at the northern stations (average $b(440)/a(440)$ of 9.35 and 8.50 for the surface and subsurface categories,

respectively), followed by the dense bottom layer (average $b(440)/a(440)$ of 8.37) and lowest values were found for the southern stations (average $b(440)/a(440)$ of 7.16 and 7.20 for the surface and subsurface categories, respectively). This correlates very well with the importance of non-algal particle absorption as shown on the beige curves on Figure 10 and their values range from 0.0055 m$^{-1}$ for Southern surface water to 0.0369 m$^{-1}$ for the bottom dense water. The green curves confirm the evidence of a subsurface algae bloom with highest phytoplankton contribution for the subsurface layer (max. 0.063 m$^{-1}$)

and lowest for the bottom water (0.006 m$^{-1}$). The CDOM had the most stable contribution across the five different categories, with average values of $a_{CDOM}440$ ranging between 0.028–0.043 m$^{-1}$.

Relative contributions of the three optically active constituents to the non-water absorption are presented in Table 2 for three specific wavelengths (443nm, 550nm and 670nm). For phytoplankton, a minimum (resp. maximum) contribution of 6%

(resp. 82%) was found for the bottom water at 550nm (resp. the southern subsurface water at 670nm). CDOM contribution ranged from 6% for southern subsurface water at 670nm to 64% for southern surface water at 443nm. Non-algal particle contribution to the non-water absorption had values ranging between 8% for the southern subsurface water at 443nm to 70% for bottom water at 670nm. As it could also be assessed qualitatively from Figure 10, this also highlights the strong dependence of the absorption budget to the wavelength of the radiation, the CDOM contribution dramatically decreasing

with increasing wavelength. The relative CDOM contribution was thus higher than what has been found in the Atlantic water in the West Spitsbergen Current (WSC) with average CDOM contribution of 42% at 412nm (Kowalczuk et al., 2019), as well as in AW north of Svalbard with CDOM contribution of 43% at 443nm (Kowalczuk et al., 2017). However it remains significantly lower than what has been found in the central and Eastern Arctic (Gonçalves-Araujo et al., 2018) with 85% of the non-water water absorption attributed to CDOM at 443nm.






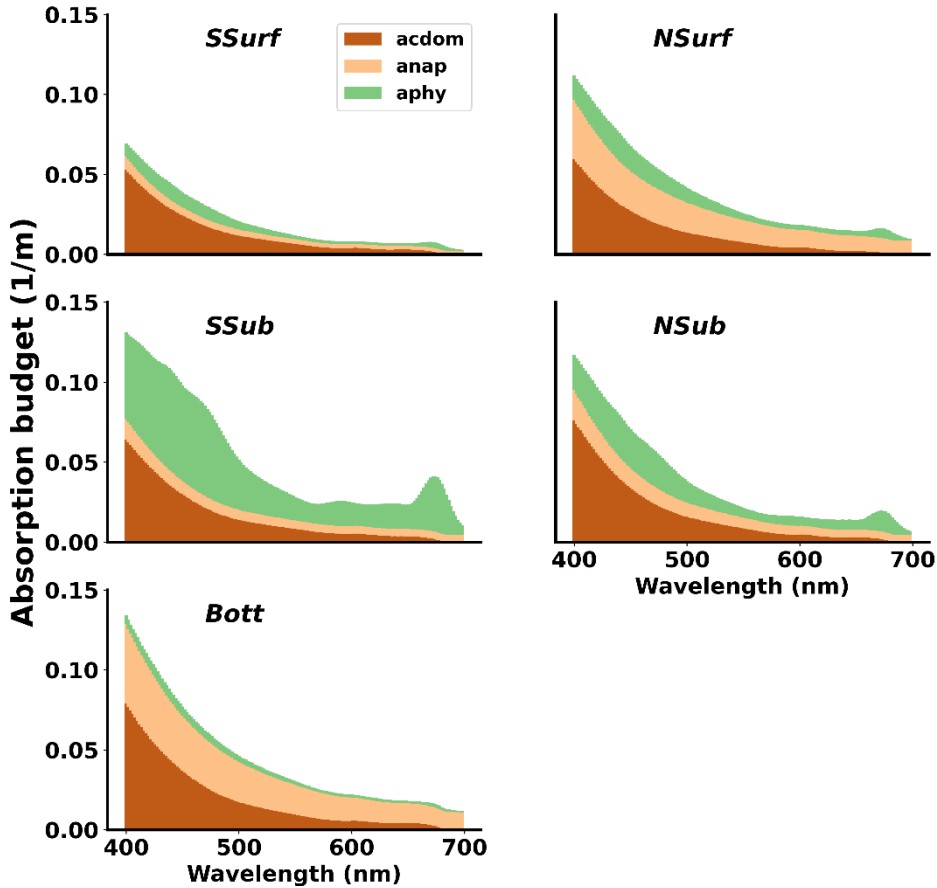

**Figure 10 – Absorption budget for five typical cases, corresponding to: (i) Northern Surface (NSurf), (ii) Southern Surface (SSurf), (iii) Northern Subsurface (NSub), (iv) Southern Subsurface (SSub) and (v) bottom dense water (Bott). Surface (resp. subsurface) water is defined as <15 m (resp. 15–60 m) water depth. The bottom dense water corresponds to the layer trapped by the sill at stations 505, 508 and 509.**





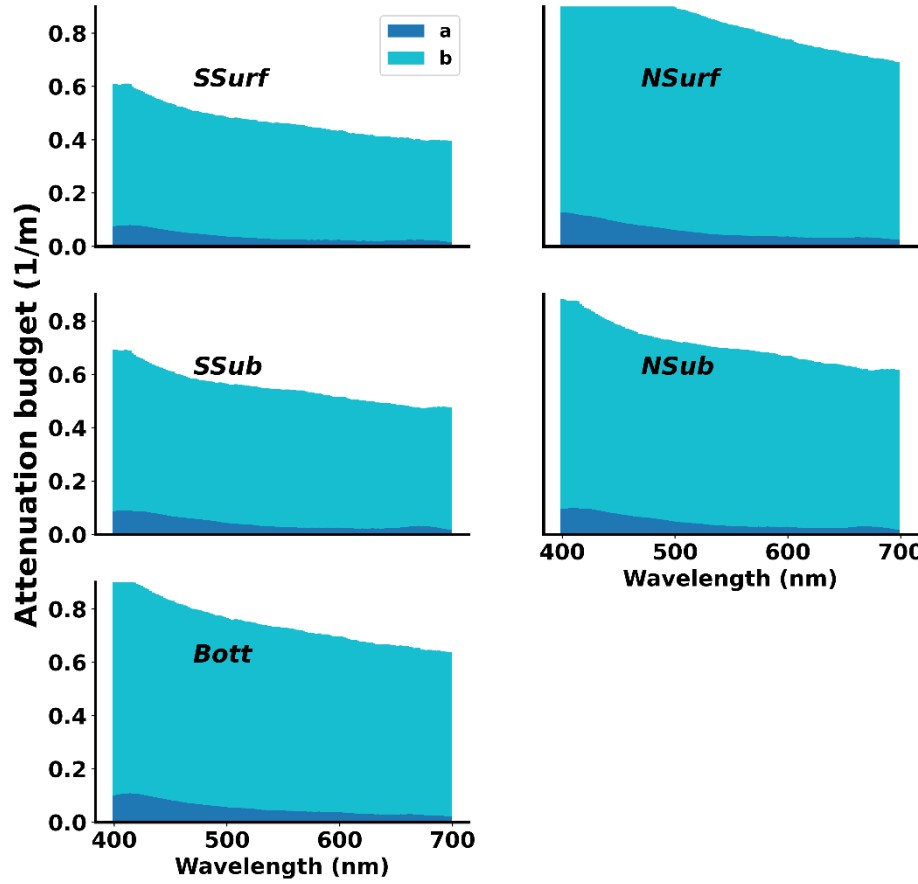

**Figure 11 - Attenuation budget for five typical cases, corresponding to: (i) Northern Surface (NSurf), (ii) Southern Surface (SSurf), (iii) Northern Subsurface (NSub), (iv) Southern Subsurface (SSub) and (v) bottom dense water (Bott). Surface (resp. subsurface) water is defined as <15 m (resp. 15–60 m) water depth. The bottom dense water corresponds to the layer trapped by the sill at stations 505, 508 and 509. Dark blue corresponds to absorption (a) and light blue to scattering (b).**

**Table 1 - Average values of the main water IOPs for the five cases corresponding to Northern Surface (NSurf), Southern Surface (SSurf), Northern Subsurface (NSub), Southern Subsurface (SSub), Dense bottom water (Bott). The number of samples n is given for the water sample-based quantities and does not apply to $b_{440}/a_{440}$ which comes from the ac-s data.**

| | $n$ | $a_{cdom}440$ (m⁻¹) | $S_{cdom}350-550$ (nm⁻¹) | $a_{NAP}440$ (m⁻¹) | $S_{NAP}350-550$ (nm⁻¹) | $a_{phy}440$ (m⁻¹) | $b440/a440$ (No unit) |
|---|---|---|---|---|---|---|---|
| *NSurf* | 6 | 0.0318 | 0.0156 | 0.0268 | 0.0071 | 0.0173 | 9.35 |
| *SSurf* | 10 | 0.0285 | 0.0157 | 0.0055 | 0.0079 | 0.0102 | 7.16 |
| *NSub* | 6 | 0.0402 | 0.0152 | 0.0135 | 0.0075 | 0.026 | 8.50 |
| *SSub* | 10 | 0.0352 | 0.0148 | 0.0091 | 0.0069 | 0.0633 | 7.20 |
| *Bott* | 3 | 0.0433 | 0.0168 | 0.0369 | 0.0067 | 0.0061 | 8.37 |



**Table 2 - Relative contributions (in %) of the phytoplankton, CDOM and non-algal particles (NAP) to the absorption budgets at 443nm, 550nm and 670nm for the five cases corresponding to Northern Surface (NSurf), Southern Surface (SSurf), Northern Subsurface (NSub), Southern Subsurface (SSub), Dense bottom water (Bott).**

| | n | *443nm* | | | *550nm* | | | *670nm* | | |
| | | *phyto* | *CDOM* | *NAP* | *phyto* | *CDOM* | *NAP* | *phyto* | *CDOM* | *NAP* |
|---|---|---|---|---|---|---|---|---|---|---|
| *NSurf* | 6 | 22 | **42** | 36 | 17 | 29 | **54** | 36 | 8 | **56** |
| *SSurf* | 10 | 24 | **64** | 12 | 20 | **55** | 25 | **45** | 26 | 29 |
| *NSub* | 6 | 33 | **50** | 17 | 29 | **40** | 30 | **64** | 11 | 25 |
| *SSub* | 10 | **59** | 33 | 8 | **52** | 29 | 19 | **82** | 6 | 12 |
| *Bott* | 3 | 7 | **50** | 43 | 6 | 33 | **61** | 12 | 18 | **70** |

## 4 Conclusion

This first study of the Inherent Optical Properties (IOP) in Storfjorden (Svalbard) highlighted its optical complexity, that we linked to the variable influence from sea-ice melt (surface stratification) and formation (dense cold bottom waters), runoff from nearby land, and phytoplankton productivity. As observed earlier in the Barents Sea proper and Atlantic waters west and north of Spitsbergen, the contribution of CDOM to the absorption budget varied quite little. Also, the contribution from humic-like FDOM was low and invariable, which indicated limited terrestrial contribution to the DOM pool in the fjord Conversely, the tryptophane-like FDOM closely followed the spatial and vertical distributions of optical proxies for chlorophyll-a concentration. Nevertheless, in surface waters not influenced by land runoff, CDOM was the main contributor to the non-water light absorption, despite its relatively low concentrations in Atlantic waters, since stratification resulted in a subsurface phytoplankton bloom with maximum magnitude between 25–50 m depth. We surmise that in shallow nearshore waters with more direct input from land runoff, the contribution from turbid plumes and non-algal particles is much larger during spring freshet and was partly found in our dataset as increased and non-negligible level of non-algal particle absorption as well as very high scattering to absorption ratios. The dense and cold bottom water, originating from winter sea ice formation and brine rejection, was found to contain higher levels of dissolved organic and non-algal particulate matter. Its transport across the sill needs further attention to better understand the potential implications regarding the bottom waters of the adjacent basins and material transports. The statistics obtained on the different IOPs and on their relative contribution to the light absorption can be used, in conjunction with the ones from previous studies in nearby areas, for improved regional parameterizations of bio-optical models used in the field of remote sensing or climate modelling.

*Data availability.* The hydrographic and optical data (Petit et al., 2021) is available at https://doi.org/10.21334/npolar.2022.e6974f73. Its processing was supported by the Nansen Legacy project.



*Author contribution.* BH and MG designed the study. HS and TP conducted the sample collection and cast profiling. RR and TP performed the sample analysis. TP, MG and RR conducted the data analysis. TP and MG wrote a first draft of the manuscript. All authors contributed to editing the manuscript.

*Competing interests.* The authors declare that they have no conflict of interest.


*Acknowledgments.* This work was funded by the Research Council of Norway through the Nansen Legacy project (RCN project no 276730). Many thanks to Hanne Sagen for supporting our participation in the UAK2020 cruise. We acknowledge the support from KV *Svalbard* crew, students and other participants of UAK2020, especially Nil Eryilmaz (UiB), Malin Lund (UiB) and Emilia Botnen van den Bergh (HVL) for their enthusiast help with water filtrations. We also warmly thank

Murat Ardelan (NTNU) for lending us the LWCC system used for this study.

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
