# Peer review of "Inherent optical properties of dissolved and particulate matter in an Arctic fjord (Storfjorden, Svalbard) in early summer"

_Ocean Science, 2021_

## Referee Comment (RC2)

**Technical comments**

There is a large number of figures and unfortunately not always placed close by the text where it is mentioned. This makes it sometimes a bit difficult to follow numbers and patterns described in the text.

Please, decide on one spelling of sea ice/ sea-ice. Mainly sea ice is used, although sea-ice is written in some lines: 14, 24, 218, 269, 289 and 395.

Line 20: An abbreviation (CDOM) is used before clearly defined. I recommend to not use abbreviations in the abstract.

Line 25: Can you please explain/ describe the pathway of dense water outflow in more details in order to reach Fram Strait? See comment below (line 65 – 67).

Line 24 - 26: It would be easier to read and understand if you write two sentences here: "Lastly, … into Fram Strait. It was found … matter likely caused by …"

Line 30: Is it Spitsbergen or Svalbard? In Line 67 your write Svalbard and not Spitsbergen.

Line 31+227: winter sea ice formation/ production

Line 50: "sparse nature" it is clear what is meant here, nevertheless, it is an uncommon wording. Is there any other possibility to make clear that field observations are rare?

Line 55: e.g. instead of eg. (two times in this line)

Line 64: distinct instead of distinctly

Line 65 – 67: From description it is hard to understand how circulation happens here. It would be helpful for understanding water mass sources for Storfjorden, if a map would be available with currents, entering/leaving path ways, Storfjorden polynya and water mass origins as well as outflows especially dense bottom water. Please add a reference for water mass definitions and currents.

Line 79: wrong citation type, brackets are wrongly placed: (see e.g. Bensi et al. (2019))

Line 104: It would be more precise if you write "In situ measurements and sampling procedure"

Line 116 + 147: Please be consistent: I recommend to put listing (i, ii, iii) italic

Line 117: Although Mili-Q is a common word when working in the laboratory.It is recommend to use the word ultra-pure water within a publication.

Line 120: Do you have any solution how to avoid variability in blanks during cruise to create blanks under the same conditions as samples are taken?

Line 136 – 139: Do you obtain water samples with the rosette separately to the IOPs or within the same cast? It seems to be a separate cast, please, make this clearer in the paragraph about water sampling (Line 136 – 139). What about depth/pressure information?

Line 148: Be careful to keep the minus at the related number together and place a space between number and unit (-80 °C) and (Line 186) be careful and keep unit together m$^{-1}$

Line 152: "…, close carefully and sealed the caps with parafilm."

Line 164: Can you please specify which three spectra are averaged?

Line 202: Can you please shortly explain "6S radiative transfer model"?

Line 210: You came up with an abbreviation not explained before: chl-a

Line 213, 219, 231, 242, 273, 321: Please, be consistent: decide for one description of the transect from north to south: upper- or lower-case letter and also the direction (north-south/ south-north)

Line 217: You came up with an abbreviation WSC, it is explained on paragraph later.

Line 219: Which station are meant with "middle of the fjord"?

Line 220: The northern stations where mentioned, if you have numbers in the figures, you can easier reference to this and it is easier to follow while reading the text.

Line 227-229: You are repeating in different words the sentence of line 217 – 219. Is it possible to combine these two sentences?

Line 261-262: Can you please give clear numbers for lowest attenuation and scattering?

Line 267-269: Please refer in line 267-269 again to the corresponding figure.

Line 278 + 289: Please add to Figure 7 which panel it meant.

Line 285: Please add related figure for $a_{CDOM}(440)$ data

Line 286+287: Brackets after citation are missing.

Line 293: Miss-spelling: "observations east of Storfjorden"

Line 305, 314, 322, 330: Stick to one type of referring to figure panels (top, middle, bottom or a, b, c).

Line 307: Can you please specify stations numbers to give an easy and quick possibility to find mentioned maximum in the figure?

Line 311: Please describe distribution/situation with a few words before referring that it looks like in the WSC.

Line 313: chl-a: except of Line 210 always phytoplankton is used. If only chlorophyll is meant that please write the full word or give explanation of abbreviation chl-a beforehand.

Line 320: Can you please indicate at which figure I can see the mentioned increase, since it is not in figure 9 which was just mentioned before.

Line 327: Brackets for citation are wrong.

Line 330: Formatting of aLH(676) is different than in the first paragraph of section 3.5.

Line 355-361: Can you give any error estimation for water-sample based values? I guess, 3 samples are not much to define significance.

Line 364-366, 371, 372, 374: Please place a space between number and unit.

Line 370: Abbreviations were already introduced for Atlantic Water (AW) and West Spitsbergen Current (WSC) so please use them.

Line 372: "However, …"

Table 2: Can you please explain what bold numbers mean within the table?

Figure 1: Please put grid below location numbers to not cover numbers.

Figure 3-8: It would be useful to have station numbers within the plots. Then it would be easier to mention station numbers within the text (see comment to Line 220).

Figure 3: It is a bit confusing presenting CTD-data and do not explain which cast/instrument set up was used to obtain presented data. It seems that you present data from the instrument package where station 511 was not sampled due to battery failure (offline measurements). Isn't it possible and better to use CTD-data from the casts made with the online CTD, when also delta 18oxygen samples where performed? Otherwise please explain why you used data with station gap instead of "full" data sets. Averaging in Ocean data view would be more precise with more stations.

Figure 4: What about combining figure 4 with figure 3. This would reduce number of figures and all parameters described in paragraph 3.1 are at one view.

Figure 5, left: Is there a legend available? In the caption: Can you please make sure that numbers and corresponding units stays together (Line 247) and additionally please add by which symbol sampling stations are indicated (green dots).

Figure 7: I suggest to add density lines to figure 7a instead of figure 6a or to both, if required. For me it seems that in figure 6 it is not used for explaining pattern within the text belonging to figure 6. For figure 7 you describe a clear pattern related to density pattern.

Figure 7+8: Please shrink figures to fit the page, so that caption fits to the same page.

Figure 8b: You use DIVA gridding for a_phy(440). Did you use the same for all other panels? I do not recommend using DIVA gridding, since it can create artificial pattern which might not exists. This makes it dangerous to interpret pattern. Please comment your decision of DIVA-gridding. In the caption it is written that density anomaly lines are presented, actually they are not visible. Please describe white points within the caption.

Figure 11: Colours are very hard to distinguish, please choose as second colour that stands out more.

---

## Author Response (AR1)

**Response to reviews - os-2021-90 -** "Inherent optical properties and optical characteristics of dissolved organic and particulate matter in an Arctic fjord (Storfjorden, Svalbard) in early summer" by Tristan Petit et al., Ocean Sci. Discuss., https://doi.org/10.5194/os-2021-90-RC1, 2021

The comments from the Referee are in black text, while our responses are in blue text.
Note that the page, figure or line numbers here denote those in the original submission, unless otherwise stated.

**Anonymous Referee #1**

The article, Inherent optical properties and optical characteristics of dissolved organic and particulate matter in an Arctic fjord (Storfjorden, Svalbard) in early summer, by Tristan Petit et al., presents a dataset acquired in summer 2020, consisting of inherent optical properties (IOPs) such as absorption, attenuation, and fluorescence of optically active constituents, both dissolved and particulate. As the authors suggest the IOPs are crucial in developing bio-optical models. The dataset does include state-of-the-art bio-optical data, which could be very useful to the scientific community involved in ocean color studies in the Arctic. However, the manuscript needs to be improved and so authors are requested to consider the following comments and suggestions.

We like to thank Referee #1 for the thorough reading of the manuscript, and the multiple comments and suggestions that have helped to improve the presentation of the manuscript. Specifically we have attempted to improve the figures by editing and reorganising them as well as by adding some new content (eg. all FDOM channels are now presented as section plots). We have also modified the title of the manuscript, and added some more discussion on the relationship to salinity, and added the suggested additional references provided by Referee 1. Detailed responses to these suggestions are given below.

Out of the total 127 comments in the annotated manuscript (attached to this response with our responses to every comment), we have followed nearly all suggestions. Regarding the few cases for which we have not fully followed Referee 1 suggestions, we have detailed the rationale behind our choices either in the annotated manuscript and/or below with our responses in this letter.

**Specific comments from Referee #1**

Any particular reason why water samples were not collected to quantify phytoplankton pigments like chla, it being a widely studied optically active constituent?
Unfortunately this expedition was part of an educational cruise, with focus on physical oceanography, and there was limited laboratory space to support such work in an efficient manner since this was a coast guard vessel and not a research vessel. Thus this was more of a ship-of-opportunity for our work, to study an area with little to no optical data. Thus we unfortunately had to opt to collect particulate abs samples at the expense of Chla samples, although in hindsight this was an omission we should have not done. However the in situ absorption measurements (absorption line height at 676 nm) are known to be a good proxy for phytoplankton biomass, also seen from the relation to the measured Chla fluorescence.

The methods section lacks references, please cite appropriate references throughout the section.

We have added a number of references where appropriate. In places we have simply written out the detailed protocol from the external labs that were used for analysis, and there is not necessarily a suitable published reference to use, since they are established laboratory practises used over long periods of time without a proper scientific reference, nevertheless proven reliable.

Can oxygen isotope values be used to quantify Dissolved Oxygen? What more can we interpret from the Oxygen isotope values?
Oxygen isotope measurements quantify the relative amount of oxygen isotopes in the water molecule, not the concentration of dissolved oxygen gas. These are two very different things. DeltaO18 is widely used in oceanography to interpret the origin of freshwater (from sea ice melt or river/glacial runoff and precipitation in the Arctic (e.g. Östlund & Hut, 1984, https://doi.org/10.1029/JC089iC04p06373), since the two freshwaters concerned (runoff or sea ice melt) have nearly the same salinity, but very different DeltaO18 signals. Thus DeltaO18 can be used to estimate the contribution from each. Here we use it to interpret the likely contribution from runoff or sea ice melting to the surface layer. We do this qualitatively, since this is the only oxygen isotope data from the fjord, to our knowledge, and it does not allow us to make quantitative estimates since we lack proper information on the end-members (sea ice and runoff) to do that properly (because more samples from Svalbard runoff and sea ice are required for that, but does not exist).

The purpose of using satellite image in the results and discussion section is unclear as there is no validation of data using field observations. I suggest to include figure 5 in section 2.4.
Our purpose was not to validate satellite products, but to show the broad aspect of the 'ocean color' situation from space at the time of sampling to place our observations in geographical context. We have now moved section 3.2 with former Figure 5 to section 3.1, right after the satellite data processing is introduced (section 2.4). This improves the flow of the text, and provides useful background information to place the in-situ observations in context. We have also slightly modified the text in former section 3.2 (now section 3.1) to better reflect the attempted use of the satellite data in the paper. Satellite validation is out of scope for this paper and given we do not have direct Chla measurements at the surface it is neither plausible to go ahead with such work, the novelty lies in the gathered in situ data in the fjord.

Investigate the relationship between surface acdom440 and salinity
Thank you for this suggestion. Indeed this should be one of the most basic approaches when dealing with oceanographic data. We have only few data of acdom440 from direct measurements (CDOM absorption samples), thus given the sensitivity of fluorescence we have here examined the relationship between salinity and FDOM data that was acquired simultaneously with the same frequency from in situ profiles. Note that we have chosen the FDOM1 and FDOM2 channels as their section plots (added in Figure 7 in the revised manuscript) do show similar patterns as the acdom440 section plot and are those channels of the FDOM instrument that are therefore most "representative" of acdom440.

[Figure]

This indeed shows a relationship with salinity, but appears inverse to that found in many typical estuaries (as now noted in the ms.). With decreasing fluorescent DOM with decreasing salinity, indicative of the role of low-CDOM sea ice melt diluting CDOM concentration at the surface, with little sign of impact of land runoff that typically is the high-CDOM source. One has to note the fairly low values of humic-like fluorescence leading to resolution-based gridding-like artefacts visible on the plots. We would prefer not to add these scatter plots within the manuscript for sake of brevity. Instead we have added this discussion in a concise way in the first paragraph of Section 3.4.

**Further references:**

Mascarenhas VJ and Zielinski O (2019) Hydrography-Driven Optical Domains in the Vaigat-Disko Bay and Godthabsfjord: Effects of Glacial Meltwater Discharge.Front. Mar. Sci. 6:335. doi: 10.3389/fmars.2019.00335, figure 4, panels C, G)
Thank you, we have added this reference in the section 3.4 where we show and discuss CDOM results

Bowers, D., and Brett, H. (2008). The relationship between CDOM and salinity in estuaries: an analytical and graphical solution. J. Mar. Syst. 73, 1–7. doi:10.1016/j.jmarsys.2007.07.001,
Thank you, we have added this reference in the section 3.4 where we show and discuss CDOM results, in relation to salinity.

Linkages between the circulation and distribution of dissolved organic matter in the White Sea, Arctic Ocean. Cont. Shelf Res. 119,1–13. doi: 10.1016/j.csr.2016.03.004, Figure 7b
Thank you, we have added this reference in the section 3.4 where we show and discuss CDOM results.

Add a table with list of acronyms and abbreviations (table 1) in section 2, rename the other tables accordingly.

Thank you, we have added a table (now Table 1) with the main acronyms used in this manuscript. Whether it should be in the main text, or supplementary material can be decided by the Editor(?).

**Figures need to be revised and rearranged. Detailed comments are added in the attached manuscript.**

Thank you for the suggestions regarding improvements to our figures. We have extracted below some of the detailed comments on the figures from the annotated manuscript, and responded to the suggested changes in more detail below.

Note - the Figure numbers here refer to the figure numbers in the original manuscript unless otherwise stated.

Figure 1 -
-show the water masses and direction of currents in the map.
-use a different color to mark the stations sampled across the fjord.
-mark the sill.
We have added arrows to indicate the major nearsurface currents in the area, based on the Fossile et al paper (which is based on earlier work in the area).
We have edited the colors of the station markers as suggested, with one color for the main South-North transect, and another for the other stations near the sill area.
We have added an arrow to indicate the sill.
We do not quite understand how we can show water masses on a map, these are on multiple depths and vary a lot geographically, which is why we do not want to indicate this due to the certain uncertainty in defining water masses.

Figure 2 -
 - add a panel b, with a second station, may be stn. 509
We have added a second panel with stn. 509

Figure 3
-reduce ODV figures to half the size
-combine figures 3 and 4
-add potential density anomaly as a separate figure, so create 4 sub figures, and arrange 2x2
-label the figures as a,b,c,d
-y axis is pressure or depth?
-follow same color for contour lines (either black or white)
-mark stations nos. over vertical lines
-edit text for figure nos. accordingly

A big reduction in the figure size would make it too small to see the needed details. We opt to slightly reduce the size of the figures.
Combining Fig 3 and 4 is a good idea. Thank you.
Potential density (or other oceanographic variables) are commonly shown in same panels in oceanographic papers, and we opt to keep this as is, due to i) easier direct comparison of the data in the same panel, ii) we are not adding more panels to the figures, which simply takes more space.
We have labeled panels with a, b, c, .. as suggested.
As the y-axis says, this is Pressure (decibar), which is widely used with oceanographic data.
We opt to use black contours when given for the same variable as in the colored contour, and in white if an additional parameter is shown as overlay (in this case it is potential density).

We have marked station numbers.

Figure 4
Merged with Figure 3 - as suggested (see above).

Figure 5
- color code the two transects as suggested in figure 1
We have now moved this figure to section 3.1 (in response to the suggestions of reviewer), and also edited the figure to be consistent with the map in Figure 1 (marker color).

Figure 6
-reduce ODV figures to half the size
-y axis is pressure or depth?
-mark stations nos. over vertical lines
A big resizing does not make sense in our opinion and we have decided to keep the same width as the (slightly reduced) Figure 3 for all the section plots of the revised manuscript.
The y-axis corresponds to pressure as is indicated on the axis label.

Figure 7
-reduce ODV figures to half the size
-y axis is pressure or depth?
- Mark stations nos. over vertical lines
- Add FDOM1 and FDOM2 section plots as supplementary material
-investigate the relationship between surface acdom440 and salinity, and add a 4th figure panel
-label figure panels as a, b, c, d
A big resizing does not make sense in our opinion and we have decided to keep the same width as the (slightly reduced) Figure 3 for all the section plots of the revised manuscript.
The y-axis corresponds to pressure as indicated with axis label..
We have marked station numbers.
We have decided to show all the FDOM channels within the result section as FDOM1 and FDOM2 do provide some information (similar patterns between FDOM1 and acdom440) even if they show low values. Thus, we have split Figure 7 into two figures: a first one with CDOM products and a second one with FDOM products.
We have now labelled each panel with a dedicated letter.

Figure 8
Size slightly reduced

Figure 9
Added a and b to the panels.

Figure 10
-label figures as a, b, c, d, d, e. be consistent with other figures
-add a sixth figure (f) with SCM depth, use the same in discussion
We have not added a sixth panel with SCM, since we did not target our sampling to the exact depth of the maximum SCM, but used fixed depths, thus the subsurface samples here represent the SCM layer with higher biomass (which is not always exactly at the maximum SCM depth), but due to fixed depth sampling, and its misleading to call it SCM in our opinion.
Labelling done as suggested, caption edited accordingly.

Figure 11 -

-label figures as a, b, c, d, d, e. be consistent with other figures

-add a sixth figure (f) with SCM depth, use the same in discussion

Same as for Figure 10

Panels labeled

**Response to reviews - os-2021-90 -** "Inherent optical properties and optical characteristics of dissolved organic and particulate matter in an Arctic fjord (Storfjorden, Svalbard) in early summer" by Tristan Petit et al., Ocean Sci. Discuss., https://doi.org/10.5194/os-2021-90-RC2, 2021

The comments from the Referee are in black text, while our responses are in blue text.

**Anonymous Referee #2**

In general, the preprint Inherent optical properties and optical characteristics of dissolved organic and particulate matter in an Arctic fjord (Storfjorden, Svalbard) in early summer by Petit et al. presents a very good picture of the IOPs and the optical complexity within the Storfjorden. Dataset enhances bio-optical picture of Arctic Ocean with state-ofthe-art data and methods. The dataset can be used to increase the amount of data to be used for modelling bio-optical complex waters in the Arctic Ocean. The preprint needs improvement, please take into consideration suggestions and comments in the attached PDF.

We thank Referee #2 for the careful reading and relevant comments and suggestions that helped improving our manuscript. We have handled and responded to all of these, either below or in the attached supplemental pdf with our responses as additional comments to the referees original comments. Overall we have made nearly all of the suggested changes by Referee #2.

**Selected specific comments (the responses to additional comments are in the attached annotated pdf):**

Do you have any solution how to avoid variability in blanks during cruise to create blanks under the same conditions as samples are taken?
Such a solution would indeed be very helpful. However it seems very hard to obtain during cruise as much control and stability on eg. air temperature and purity of reference water as is the case in a laboratory. That is why there is no consensus regarding the choice of using the blanks measured during cruise (close temporal match with cruise data but more variability) or right before/after cruise in a lab (not as close temporal match but less variability).

You describe the ratio of scattering to absorption. Can you please discus this ratio in more details?
Thank you for this suggestion. The corresponding paragraph in the section 3.3 was indeed lacking some discussion about the link between scattering to absorption ratio and particle types. We have thus added some discussion in the revised manuscript on how this ratio can help in interpretation of the particulate material composition observed.

You clearly describe spectral slope. Can you please discuss this parameter in more details?
Thank you for this suggestion. We have added some more discussion and useful references about CDOM spectral slope as an indicator of DOM in section 3.4 of the revised manuscript.